

# Mitigation of block withholding attack based on zero-determinant strategy

Min Ren[1], Hongfeng Guo[1] and Zhihao Wang[2]

[1] School of Statistics and Mathematics, Shandong University of Finance and Economics, Jinan, Shandong Province, China
[2] School of Management and Engineering, Shandong University of Finance and Economics, Jinan, Shandong Province, China

## ABSTRACT

This article focuses on the mining dilemma of block withholding attack between the mining pools in the bitcoin system. In order to obtain the higher revenue, the rational mining pool usually chooses an infiltration attack, that is, the pool will falls into the mining dilemma of the PoW consensus algorithm. Thus the article proposes to apply zero-determinant strategies for optimizing the behavior selection of the mining pool under PoW consensus mechanism to increase the total revenues of the system, so as to solve the mining dilemma. After theoretically studying the set and extortionate strategy of zero-determinant, the article devises an adaptive zero-determinant strategy that the pool can change the corporation probability of the next round based on its previous revenues. To verify the effectiveness of zero-determinant strategies, based on the actual revenue of the mining pool defined and deduced in the paper, it simulates 30 sets of game strategies to illustrate the revenue variation of the mining pools. The simulation results show that the three zero-determinant strategies can effectively improve the convergence rate of cooperation, mitigate block withholding attack and maximize the total revenues of the system. Compared with the set and extortionate strategy, the adaptive strategy can ensure more stability and more revenue.

## INTRODUCTION

As a decentralized shared ledger, the blockchain ensures the non-tampering property and unforgeability of transactions with an asymmetric encryption algorithm, realizes decentralization through the peer-to-peer (P2P) technology of point-to-point self-organizing network, and guarantees the consistency of block data between nodes, using a consensus algorithm (*Nakamoto, 2009*). Due to its special properties, the blockchain has been widely used in many fields (*Ren et al., 2021*, *2022*). Bitcoin is one of the most successful applications of the blockchain. It introduces the proof of work (PoW) mechanism to the block generation process. In the bitcoin system, every node participates in the production of blocks, and provides the PoW. The node that produces a block faster than others will receive a bitcoin reward. Here, the block generation is called mining, and the mining nodes are known as miners (*Rosenfeld, 2011*). Currently, each miner can

Corresponding author
Min Ren, rm_sd@163.com

receive a reward of 12.5 bitcoins (BTC) for unearthing a block, and the reward is halved every 4 years. On average, it takes about 10 min to produce a block. The difficulty in mining is adjusted automatically by the system every 2 weeks. The growing difficulty of mining means a miner needs to spend a long time before receiving a revenue. To obtain stable, higher income, miners choose to work cooperatively in open mining pools. Each mining pool consists of an administrator and several miners. The miners continue to send partial or complete PoWs to the administrator, who will distribute the revenue to the miners according to their shares of the workload. Most mining pools are open to the public. Any miner can join such a mining pool by providing a public network interface. As a result, open mining pools are highly susceptible to attacks.

To gain more revenue, some mining pools send their own miners to infiltrate other pools. These miners only send partial PoWs to the administrator, and discard the complete PoW being acquired. In other words, the miners receive the partial revenue from the infiltrated pool, without contributing effective computing power. This behavior is called a block withholding (BWH) attack (*Courtois & Bahack, 2014*; *Bag, Ruj & Sakurai, 2017*). Rather than provide effective revenue to the pool being attacked, the BWH attacker shares the revenue of the pool, such that the attacked pool receives less revenue and the attacking pool losses computing power. When all pools attack each other, their overall revenue will be lower than that when no attack takes place. To gain more revenue, all mining pools with rational thinking will choose to infiltrate others, *i.e.*, fall into the mining dilemma (*Eyal, 2015*) of PoW consensus algorithm. This is equivalent to the prisoner's dilemma in the game theory (*Kenter & Meigs, 2016*; *Kostyuk, 2013*). The state (attack, attack) is the only Nash equilibrium of the miners' dilemma (*Barlow, 2014*; *Carbonell-Nicolau & McLean, 2018*). The attack is the optimal strategy for individuals, but not optimal for the system. At present, the mining pools in China account for 81% of the computing power of the bitcoin network, and joining mining pools is the most important way for miners to obtain revenue. Hence, it is an urgent task to solve the miners' dilemma

The zero-determinant (ZD) strategy is an emerging approach in the game theory. As a hybrid strategy set, the ZD strategy controls the players' strategy selection by probability. This strategy breaks through the traditional Nash equilibrium theory, and optimizes the prisoner's dilemma model (*Press & Dyson, 2012*). On the one hand, the strategy presents a solution to the low system revenue. On the other hand, a player following this strategy ensures that his/her revenue is linearly correlated with the opponent's revenue, regardless of the opponent's strategy (*Hilbe et al., 2015*). The core of this article is to utilize the ZD strategies to optimize the selection of mining pool behaviors under the PoW consensus mechanism, aiming to increase the per-capita revenue, and thus solve the mining disaster induced by the BWH attack.

The main contributions of this article are as follows. (1) Assuming that the entire network has only two mining pools and the honest miners, the article derived the calculation formulas for the actual revenue of each mining pool, when the BWH attack is launched by one or both sides. (2) The article creatively used the ZD strategies to mitigate the BWH attacks. Thus the set strategy and extortionate strategy of the ZD were investigated firstly, and then an adaptive ZD strategy was proposed, under which the

mining pools will change the cooperation probability in the next round based on the revenues in the previous rounds. The proposed adaptive strategy effectively eases the BWH attack between the pools, promotes the cooperation between the pools, and increases the overall revenues of the mining pools. (3) The revenue variation of the pools was simulated under 30 sets of game strategies to verify that the ZD strategies especially the proposed adaptive strategy can effectively mitigate the BWH attack between mining pools.

## RELATED WORKS

### Research of BWH and mitigation strategy

*Nakamoto (2009)* proposed the concept of 51% attack. The ledger of the blockchain needs to be maintained by all the nodes in the network; an attacker must master 51% of the computing power of the whole network in order to tamper with the data in the ledger, which is recognized as the first attack on bitcoin consensus mechanism. Traditionally, it is believed that the safety of bitcoin can be guaranteed, as long as the miners possessing most of the computing power remain honest. With the development of bitcoin, Finney (*Wikipedia, 2022*) suggested that an attacker can realize double spending by maliciously withholding blocks. In 2011, *Rosenfeld (2011)* formally put forward the concept of BWH attack, which indicated after joining a mining pool, the attacker only provided the partial PoW (*Kwon et al., 2017*), maliciously withheld blocks, and simultaneously harmed the revenue of him/her and that of the pool. *Courtois & Bahack (2014)* extended the BWH attack, held that an attacker can freely distribute his/her computing power between mining independently and attacking the target pool, and demonstrated that the attacker can gain relatively more reward in this scenario. *Bag, Ruj & Sakurai (2017)* presented sponsored BWH attack to account for the probability that a miner might be employed by a pool to attack other pools. In 2014, the mining pool Eligius was hit by a massive BWH attack (*Courtois & Bahack, 2014*), which brought a loss of 300 BTC. The BWH attack both harms the interests of the pools, and threatens the stability of the bitcoin network. Therefore, an effective strategy should be designed to mitigate and resist such an attack.

One mitigation approach is to improve the PoW algorithm in terms of task assignment and reward mechanism. *Rosenfeld (2011)* presented the defense mechanism of task assignment. Under the mechanism, the administrator of a pool redistributes the effective PoW to the miners for calculation; any miner failing to submit the block is deemed as an attacker. However, the miners are forced by the administrator to complete additional computing tasks, resulting in a waste of computing power. Since the miners are rewarded by the administrator, who evaluates their contributions according to partial PoWs, *Schrijvers et al. (2017)* proposed an incentive-compatible reward mechanism, which encourages miners to submit blocks immediately in exchange for reward, thereby ensuring the revenue of the pool. *Bag & Sakurai (2016)* created the incentive mechanism with extra reward, which showed a miner submitting the block received an extra reward in addition to the reward proportional to his/her contribution, while an attacker never received any extra reward. Later, *Bag, Ruj & Sakurai (2017)* developed a mitigation scheme for the BWH attack between mining pools based on hash function encryption. Under the scheme, the attack is withstood as the miners cannot differentiate between partial and complete

PoWs. Nevertheless, the attacking pool increases its revenue with the partial PoWs submitted by the infiltration miners, without needing to spend extra computing power to calculate the complete PoW. Hence, incentives do not work on the administrator of the attacking pool. Besides, the task assignment mechanism has an inherent defect, namely, the miners often carry out useless computations, resulting in a waste of computing power.

In 2015, *Eyal (2015)* explored the mining disaster induced by the BWH attack. Specifically, the game between mining pools was qualitatively analyzed under the mutual attacks between two pools and multiple pools, and treated as an iterated prisoner's dilemma (IPD). The Nash equilibrium theory was adopted to prove that the mutual attacks reduced the revenues of all pools, forcing them to converge to the closed and stable state. Hence, another mitigation strategy for the BWH attack is grounded on the prisoner's dilemma model. *Tang et al. (2017)* further investigated the pure strategy and mixed strategy problems in game dilemmas, and optimized the system revenue of single pool mining dilemma with the ZD strategy. Their strategy ensures that the revenue of attacking miners is linearly correlated with that of honest miners in the pool, increases the revenue of the entire pool, and mitigates the loss of the pool brought by the BWH attack.

## Research of ZD strategy

The ZD strategy, initially proposed by *Press & Dyson (2012)* has attracted widespread attention. *Hilbe, Traulsen & Sigmund (2015)* considered three different strategy classes, including ZD, for the Iterated Prisoner's Dilemma and characterized these three classes within the space of memory-one strategies. *Ren et al. (2014)* extend the theory of ZD strategies to multiplayer games to describe which strategies maintain cooperation and proposed two simple models of alliances in multiplayer dilemmas to show how individuals could further enhance their strategic options by coordinating their play with others. Later, *Hilbe et al. (2015)* pointed out that the ZD strategy was not evolutionarily stable in some cases, and stable in some other conditions. *He et al. (2016)* applied the ZD strategy to multi-person multi-strategy iterative game model, and proved that every player could act as the leader to control the expected revenue of his/her opponents. *Hao, Rong & Tao (2015)* proposed a general model of the ZD strategies for noisy repeated games and derived the pinning strategy under noise, by which the ZD strategy player coercively sets the opponent's expected payoff to his desired level, although his payoff control ability declines with the increase of noise strength. *Mcavoy & Hauert (2017)* applied the ZD strategy theory from traditional synchronous games to alternate games, and discussed the autocratic strategy both in a strictly-alternating game and in a randomly-alternating game. *Ueda & Toshiyuki (2020)* provided a general framework for investigating situations where more than one players employ ZD strategies in terms of linear algebra and theoretically proved general mathematical properties of the ZD strategy. *Ueda (2021)* defined and provided examples of memory-two ZD strategy in the repeated prisoner's dilemma game. *McAvoy & Hauert (2016)* introduced a broader class of autocratic strategies by extending ZD strategies to iterated games with the continuous action space. These studies have enriched the theory of ZD strategy.

In reality, the ZD strategy has been applied in various fields, for its good properties and unlimited research potential. *Daoud, Kesidis & Liebeherr (2014)* introduced the ZD strategy to the secondary sharing of licensed spectrum, likened the licensed spectrum problem to the non-cooperative iterative game model of power control, and determined the long-term mean rate by changing the power level for the maximal value, which could be realized through the ZD strategy. *Zhang et al. (2014a)* applied the ZD strategy to manage the deceptions in wireless network cooperation and to share wireless network resources (*Zhang et al., 2016*), and described the resource sharing between players with an IPD game model. Regardless of the opponent's strategy, a player could guarantee the high and stable system revenue with the ZD strategy. *Zhang et al. (2014b)* also implemented the ZD strategy to small cell networks to maximize the system revenue. *Pan et al. (2014)* incorporated the ZD strategy to public good game model, and drew the following conclusions. When the number of players or the multiplication factor was small, a player could unilaterally control the expected revenue of all the other players through the ZD strategy, and set the proportion of his/her own revenue to the overall revenue of all the other players. In the IPD game model, regardless of the opponent's strategy, the ZD strategy can control the expected revenue of the opponent, and keep it linearly correlated with the expected revenue of the player adopting the strategy. All the above studies provide a reference for our research, which tries to migrate the BWH attack with the ZD strategy.

## MINING DILEMMA ANALYSIS

### Calculation of actual revenue of mining pool

The BWH attack can be traced back to the nascency of pool mining. The attacker could be an administrator of a mining pool. He/she might arrange the computing power under his/her control to mine honestly or infiltrate another pool. The attacking behavior is either honest mining or withholding blocks. Such an attack will harm the mining revenue of the attacked pool and other participants. The attacking miners will only send partial PoWs to the attacked pool, and discard any complete PoW being acquired. The pool will continue to distribute mining revenue to the attacker, but cannot benefit from the computing power of the attacker. In this way, both the revenue of every participant of the attacked pool and that of the attacker will be lowered. However, the pool that launches the attack eyes the maximization of its own revenue. Then, whether the revenue of the attacking pool will increase or decrease through the BWH attack? To answer this question, it is necessary to define the calculation formulas for the pool revenues.

1. Pool revenue

For simplicity, the computing power is regarded as equivalent to the revenue. The greater the computing power, the stronger the competitiveness of a pool, and the more revenue acquired by the pool through mining. Obviously, the total revenue of the bitcoin system is the revenue obtained by the effective computing power of the system. The system computing power $H$ is defined as the total computing power of all miners in the bitcoin network. The effective computing power of the system is equal to the system computing power minus the computing power for the BWH attack $H_{attack}$. Similarly, the total

computing power of a pool $h_{pool}$ minus the computing power for the BWH attack $h_{attack}$ is the effective computing power of a pool. Then, the revenue of a pool can be defined as

$$R_{pool} = \frac{h_{pool} - h_{attack}}{H - H_{attack}}. \tag{1}$$

2. Average revenue of miners

The average revenue of the miners in a pool can be defined as

$$R_{miner} = \frac{R_{pool}}{h_{pool}}. \tag{2}$$

3. Honest mining revenue

The honest mining revenue of a miner is the average revenue of the miners multiplied by the computing power of honest miners. That is, the honest mining revenue is distributed according to the computing power of honest miners $h_{honest}$ as a proportion of the total computing power of the pool $h_{pool}$. The formula is

$$R_{honest} = h_{honest} \cdot \frac{R_{pool}}{h_{pool}} = h_{honest} \cdot R_{miner}. \tag{3}$$

4. Attack revenue

The attack revenue is the product of the average revenue of the miners in the attacked pool and the computing power of the attack $h_{attack}$.

$$R_{attack} = h_{attack} \cdot \frac{R_{pool}}{h_{pool}} = h_{attack} \cdot R_{miner} \tag{4}$$

5. Actual revenue

The actual revenue of a pool is the sum of the revenue of honest mining and the attack revenue.

$$R_{real} = R_{honest} + R_{attack} \tag{5}$$

The actual bitcoin network is rather complicated. There are many mining pools in the whole network. Each pool has complex mining behaviors. To facilitate the analysis of the BWH attack, this paper considers the simplest situation, *i.e.*, there are only two pools named $Pool_1$ and $Pool_2$ in the network, and the other miners conduct mining honestly and independently. Suppose the computing power of the entire network is 1, and the computing power of $Pool_1$, $Pool_2$ and the other miners are $h_1$, $h_2$ and $h_3$, respectively. It is obvious that $h_1 + h_2 + h_3 = 1$. The administrator of each pool distributes block reward fairly to each miner according to his/her proportion of computing power. If no attack takes place, the actual revenues of $Pool_1$ and $Pool_2$ are $h_1$ and $h_2$, respectively. The following is an analysis on the revenue variation of each pool under two difference scenarios, namely, the BWH attack launched by only one pool and the attack launched by both pools.

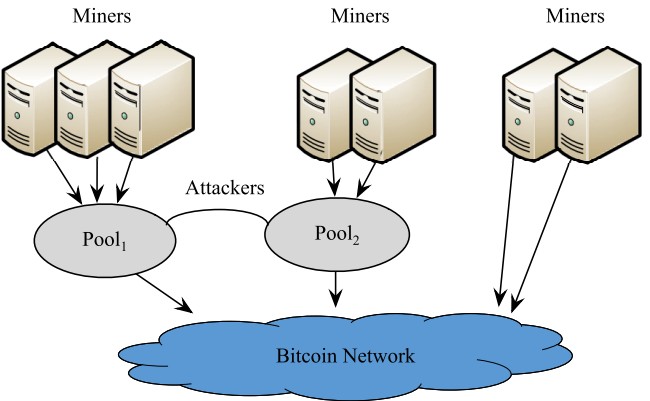

**Figure 1 *Pool₁* Attacking *Pool₂*.**

### Unilateral attack

Figure 1 shows the scenario of unilateral attack. It is assumed that $Pool_1$ attacks $Pool_2$ with $r_1 h_1$ $(0 < r_1 < 1)$ of its own computing power, and conducts honest mining with the remaining computing power $(1 - r_1)h_1$, while $Pool_2$ does not attack $Pool_1$. Note that $r_1$ is the infiltration rate, *i.e.*, the percentage of infiltration miners in all miners of the pool (Normally, $r_1 = 0.1$). Then, the effective computing power of the entire network is $1 - r_1 h_1$.

The pool revenues of $Pool_1$ and $Pool_2$ are

$$R_{pool_1} = \frac{(1 - r_1)h_1}{1 - r_1 h_1}, \tag{6}$$

$$R_{pool_2} = \frac{h_2}{1 - r_1 h_1}. \tag{7}$$

The average revenue of the miners in $Pool_1$ and $Pool_2$ are

$$R_{miner_1} = \frac{1}{1 - r_1 h_1}, \tag{8}$$

$$R_{miner_2} = \frac{h_2}{(1 - r_1 h_1)(r_1 h_1 + h_2)}. \tag{9}$$

The honest mining revenue of a miner in $Pool_1$ and $Pool_2$ are

$$R_{honest_1} = \frac{(1 - r_1)h_1}{1 - r_1 h_1}, \tag{10}$$

$$R_{honest_2} = \frac{h_2^2}{(1 - r_1 h_1)(r_1 h_1 + h_2)}. \tag{11}$$

The attack revenue of $Pool_1$ is

$$R_{attack_1} = \frac{r_1 h_1 h_2}{(1 - r_1 h_1)(r_1 h_1 + h_2)}. \tag{12}$$

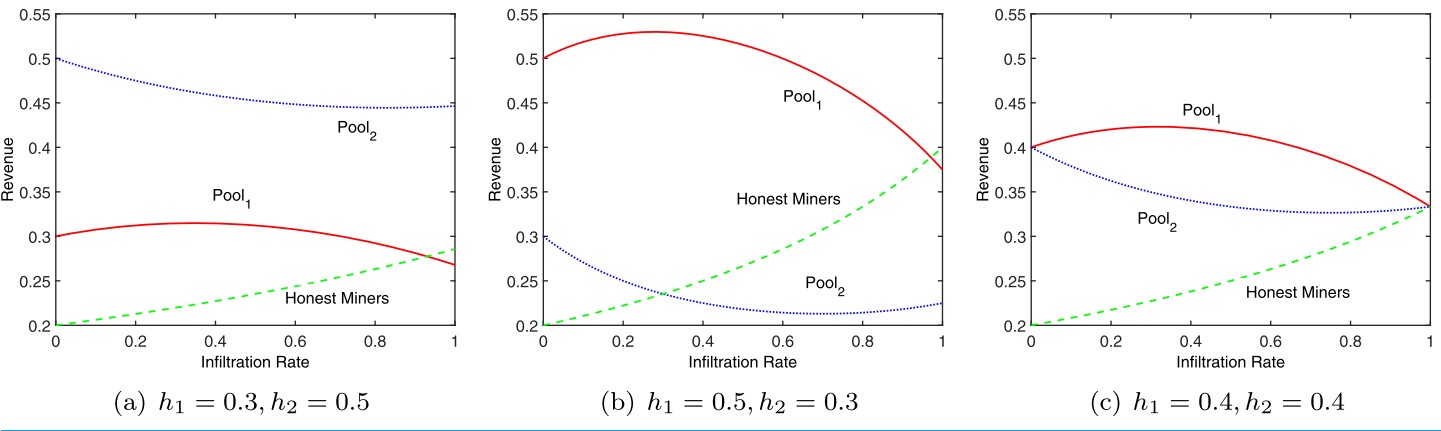

(a) $h_1 = 0.3, h_2 = 0.5$  (b) $h_1 = 0.5, h_2 = 0.3$  (c) $h_1 = 0.4, h_2 = 0.4$

**Figure 2 Revenue at different infiltration rates.**

The actual revenue of $Pool_1$ and $Pool_2$ are

$$R_{real_1} = \frac{h_1 h_2 + r_1 h_1^2 - r_1^2 h_1^2}{(1 - r_1 h_1)(r_1 h_1 + h_2)}, \tag{13}$$

$$R_{real_2} = \frac{h_2^2}{(1 - r_1 h_1)(r_1 h_1 + h_2)}. \tag{14}$$

If $Pool_1$ does not launch an attack, the original revenue of $Pool_1$ is $R'_{real1} = h_1$. Let

$$\Delta R_1 = R_{real_1} - R'_{real_1} = \frac{r_1 h_1^2 (r_1 h_1 - r_1 + h_2)}{(1 - r_1 h_1)(r_1 h_1 + h_2)}, \text{ where } \frac{r_1 h_1^2}{(1 - r_1 h_1)(r_1 h_1 + h_2)} > 0. \text{ When }$$

$r_1 < \dfrac{h_2}{1 - h_1}$, $r_1 h_1 - r_1 + h_2$ is also greater than zero, *i.e.*, $\Delta R_1 > 0$. In this case, $Pool_1$ can obtain more revenue than what it can obtain without launching any attack. Apparently, there is a suitable infiltration rate for $Pool_1$ to maximize its revenue, showed in Fig. 2.

If $Pool_1$ does not launch an attack, the original revenue of $Pool_2$ is $R'_{real2} = h_2$. Then, $\Delta R_2 = R_{real_2} - R'_{real_2} = \dfrac{r_1 h_1 h_2 (r_1 h_1 + r_1 h_2 - 1)}{(1 - r_1 h_1)(r_1 h_1 + h_2)}$. Since $r_1 h_1 + r_1 h_2 - 1 < 0$ and any other term in the formula is greater than zero, $\Delta R_2 < 0$. Hence, as shown in Fig. 2, an attack by $Pool_1$ will definitely lower the revenue of $Pool_2$. Therefore, a rational pool will choose to attack in order to control its own loss, that is, both pools will choose to attack. The total revenue of miners engaged in independent mining is $R_3 = \dfrac{h_3}{1 - r_1 h_1}$. Since $1 - r_1 h_1 < 1$, the actual revenue of these miners is greater than the original revenue $h_3$.

### Mutual attacks

Under the premise that $Pool_1$ launches an attack, it is assumed that $Pool_2$ attacks $Pool_1$ with $r_2 h_2$ ($0 < r_2 < 1$) of its computing power, as is shown in Fig. 3. In this case, the effective computing power of the entire network is $1 - r_1 h_1 - r_2 h_2$.

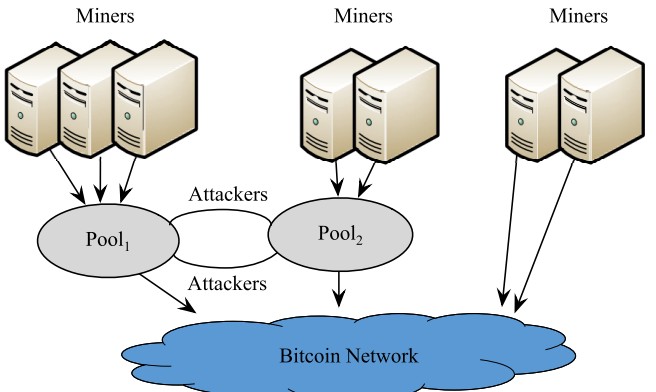

**Figure 3 Mutual attacks between the two pools.**

The pool revenues of $Pool_1$ and $Pool_2$ are

$$R_{pool_1} = \frac{(1 - r_1)h_1}{1 - r_1h_1 - r_2h_2}, \tag{15}$$

$$R_{pool_2} = \frac{(1 - r_2)h_2}{1 - r_1h_1 - r_2h_2}. \tag{16}$$

The average revenue of the miners in $Pool_1$ and $Pool_2$ are

$$R_{miner_1} = \frac{(1 - r_1)h_1}{(1 - r_1h_1 - r_2h_2)((1 - r_1)h_1 + r_2h_2)}, \tag{17}$$

$$R_{miner_2} = \frac{(1 - r_2)h_2}{(1 - r_1h_1 - r_2h_2)(r_1h_1 + (1 - r_2)h_2)}. \tag{18}$$

The honest mining revenue of a miner in $Pool_1$ and $Pool_2$ are

$$R_{honest_1} = \frac{(1 - r_1)^2 h_1^2}{(1 - r_1h_1 - r_2h_2)((1 - r_1)h_1 + r_2h_2)}, \tag{19}$$

$$R_{honest_2} = \frac{(1 - r_2)^2 h_2^2}{(1 - r_1h_1 - r_2h_2)(r_1h_1 + (1 - r_2)h_2)}. \tag{20}$$

The attack revenue of $Pool_1$ and $Pool_2$ are

$$R_{attack_1} = \frac{r_1(1 - r_2)h_1h_2}{(1 - r_1h_1 - r_2h_2)(r_1h_1 + (1 - r_2)h_2)}, \tag{21}$$

$$R_{attack_2} = \frac{(1 - r_1)r_2h_1h_2}{(1 - r_1h_1 - r_2h_2)((1 - r_1)h_1 + r_2h_2)}. \tag{22}$$

The actual revenue of $Pool_1$ and $Pool_2$ are

$$R_{real_1} = \frac{(1 - r_1)^2 h_1^2}{(1 - r_1h_1 - r_2h_2)((1 - r_1)h_1 + r_2h_2)} + \frac{r_1(1 - r_2)h_1h_2}{(1 - r_1h_1 - r_2h_2)(r_1h_1 + (1 - r_2)h_2)}, \tag{23}$$

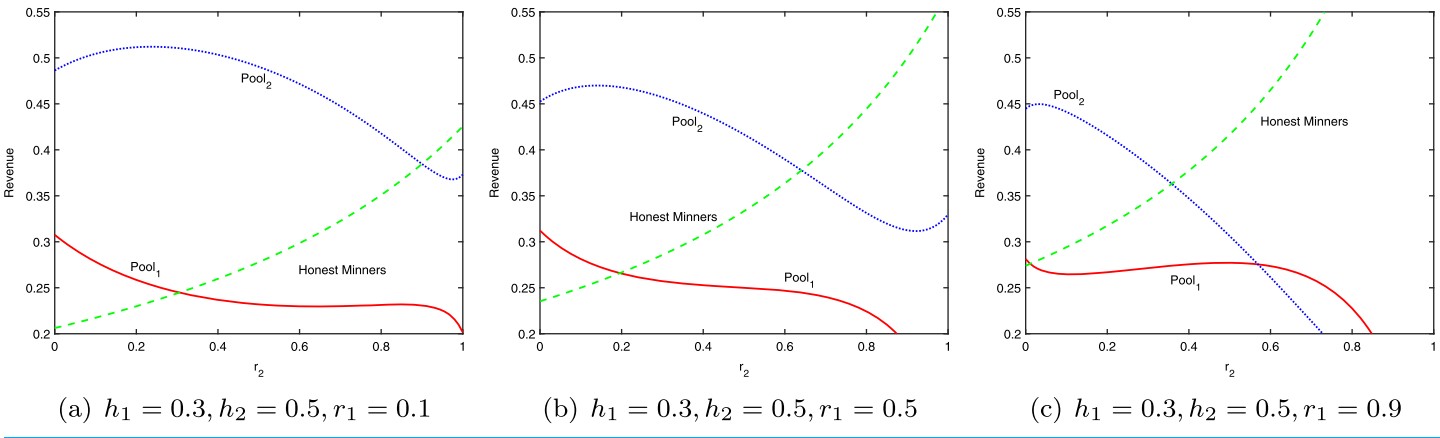

**Figure 4  Revenue at different infiltration rates.**

$$R_{real_2} = \frac{(1-r_2)^2 h_2^2}{(1-r_1 h_1 - r_2 h_2)(r_1 h_1 + (1-r_2)h_2)} + \frac{(1-r_1)r_2 h_1 h_2}{(1-r_1 h_1 - r_2 h_2)((1-r_1)h_1 + r_2 h_2)}. \quad (24)$$

Figure 4 shows the revenues of the two pools and honest miners at different infiltration rates, with $h_1 = 0.3$ and $h_2 = 0.5$. When the two pools attack each other, in most cases, the revenue of each side is lower than that of honest mining, yet higher than that under the scenario of being attacked but not attacking the other side. This phenomenon can be explained by the revenue variation of the miners engaged in independent mining. The actual total revenue of independent miners $R_3 = \dfrac{h_3}{1 - r_1 h_1 - r_2 h_2}$ is always above the original revenue $h_3$. Whereas the total revenue of the bitcoin system is fixed, the total revenue of $Pool_1$ and $Pool_2$ will definitely drop. In each round of mining, the best strategy for each pool is to attack the other side. When the system reaches the equilibrium, the revenue of each pool will be lower than that of honest mining. This is the mining dilemma, which is comparable to the classic prisoner's dilemma in the game theory.

In the entire bitcoin system, a pool administrator can select the strategy for each round of mining, *i.e.*, how much computing power should be reserved for mining in the pool, and how many miners should be sent to launch the BWH attack. From the perspective of the IPD, the continuous mining competition between pools is an iterative game. In each round, each pool, as a game party, can choose between launching an BWH attack (*i.e.*, the defection strategy of the prisoner's dilemma) and not to attack (*i.e.*, the cooperation strategy of the prisoner's dilemma).

## Prisoner's dilemma and IPD

The prisoner's dilemma, proposed by A. W. Tucker in 1950 (*Tucker & Luce, 1959*), is a classic problem in the game theory. The model involves two members *X* and *Y* of a gang of robbers, who have been arrested and interrogated in separate rooms. If both plead guilty, *i.e.*, choose defection, each will be sentenced to 3 years; if one pleads guilty and the other

**Table 1 Revenue matrix of prisoner's dilemma.**

| | | Y | |
| --- | --- | --- | --- |
| | | Corporation, C | Defection, D |
| X | Corporation, C | (R, R) | (S, T) |
| | Defection, D | (T, S) | (P, P) |

does not, the former will be released immediately, while the latter will be sentenced to 5 years; if both do not plead guilty, *i.e.*, choose cooperation, each will be sentenced to 1 year.

The revenues of the two prisoners can be described by Table 1, where $R$ is the reward for mutual cooperation; $T$ is the temptation to defect for the defector; $S$ is the sucker's payoff when one party chooses defection and the other chooses cooperation; $P$ is the punishment for mutual defection. These parameters satisfy $T > R > P > S$ and $2R > T + S$, and normally $(T, R, P, S) = (5, 3, 1, 0)$. For $X$, if $Y$ chooses cooperation, his/her best choice is defection; if $Y$ chooses defection, he/she will also choose defection, because defection reduces his/her loss. Hence, regardless of the other party's choice, the best choice is always defection. Rational prisoners will always betray each other. That is, (defection, defection) is the Nash equilibrium of the prisoner's dilemma. However, the Nash equilibrium point is not necessarily the optimal strategy combination for system revenue. The revenue of the state is below that under (cooperation, cooperation). Therefore, the prisoners face the dilemma of selecting between cooperation and defection (*Press & Dyson, 2012*).

An iterative game is multiple (greater than two) repetitions of a game. If the prisoner's dilemma only lasts one round, (defection, defection) is the inevitable outcome. In the Iterated Prisoner's Dilemma (IPD) model, however, the same gamers encounter each other repeatedly. If a player chooses to defect, he/she must consider the fact that the other side also prefers defection to reduce loss. Hence, the revenues of both sides will remain low. Then, each side of the game faces the pressure that long-term revenue is better than short-term revenue. Since the game is iterative, a rational prisoner will not stick to defection, but cautiously choose between defection and cooperation according to the selection of the opponent in the previous round. Defection might evoke punishment from the opponent, and cooperation might invite a return favor. If the iterative game lasts indefinitely, the equilibrium of (cooperation, cooperation) might appear.

## ZD STRATEGY

There is a zero-determinant strategy in IPD game, enforcing linear relationships on the payoffs. The ZD strategy is very surprising that a player can exert unilateral control over iterated interactions, regardless of his/her opponent's strategy (*McAvoy & Hauert, 2016*).

During the IPD game, a player can deduce the opponent's strategy from the game results of the previous rounds, and choose his/her strategy for the next round. It is assumed the players can only memorize a limited history. It has been proved that long-term memory is not superior to short-term memory, if the game repeats itself indefinitely, *i.e.*, the players, revenue matrix, and game strategy set are the same in each round.

**Table 2 Strategy selection probabilities of the players in the current round.**

| | | Y | |
| --- | --- | --- | --- |
| | | Corporation, C | Defection, D |
| X | Corporation, C | $p_i q_j$ | $p_i(1 - q_j)$ |
| | Defection, D | $(1 - p_i)q_j$ | $(1 - p_i)(1 - q_j)$ |

Therefore, the following analysis assumes that each player can only remember the game outcome of the previous round, *i.e.*, have only one-step memory.

For player $X$, the game result can be represented as $XY \in (CC, CD, DC, DD)$, where $C$ and $D$ are cooperation and defection, respectively. For player $Y$, the game result can be represented as $YX \in (CC, DC, CD, DD)$. The revenues of $X$ and $Y$ can be vectorized as $U_X = (R, S, T, P)$ and $U_Y = (R, T, S, P)$, respectively. Let $p = (p_1, p_2, p_3, p_4)$ and $q = (q_1, q_2, q_3, q_4)$ be the probability for $X$ and $Y$ to choose cooperation according to the four game results in the previous round, respectively. Then, the strategy selection probabilities of the players in the current round can be summarized as Table 2.

According to the game sequence of $X$, the transfer of $X$ and $Y$'s strategy selection can be described as a Markov state transfer matrix.

$$M = \begin{bmatrix} p_1 q_1 & p_1(1 - q_1) & (1 - p_1)q_1 & (1 - p_1)(1 - q_1) \\ p_2 q_3 & p_2(1 - q_3) & (1 - p_2)q_3 & (1 - p_2)(1 - q_3) \\ p_3 q_2 & p_3(1 - q_2) & (1 - p_3)q_2 & (1 - p_3)(1 - q_2) \\ p_4 q_4 & p_4(1 - q_4) & (1 - p_4)q_4 & (1 - p_4)(1 - q_4) \end{bmatrix} \tag{25}$$

There are four possible results of each round. Thus, each row of $M$ adds up to 1, *i.e.*, $M$ has a unit eigenvalue. Suppose $M' \equiv M - I$. Then, $\det(M') = 0$. The steady-state vector of $M$ can be represented as $v = (v_1, v_2, v_3, v_4)^T$, and $v_1 + v_2 + v_3 + v_4 = 1$. Any vector proportional to the steady-state vector satisfies $v^T M = v^T$, or $v^T M' = 0$.

According to Cramer's rule, $Adj(M')M' = \det(M')I = 0$. Hence, each row of $Adj(M')$ is proportional to $v$.

The dot product between $v$ and any four-dimensional vector $f$ can be represented as

$$v \cdot f \equiv D(p, q, f) = \det \begin{bmatrix} -1 + p_1 q_1 & -1 + p_1 & -1 + q_1 & f_1 \\ p_2 q_3 & -1 + p_2 & q_3 & f_2 \\ p_3 q_2 & p_3 & -1 + q_2 & f_3 \\ p_4 q_4 & p_4 & q_4 & f_4 \end{bmatrix}. \tag{26}$$

The second column of the determinant is controlled by $X$ separately.

$$\tilde{p} \equiv (-1 + p_1, -1 + p_2, p_3, p_4) \tag{27}$$

The third column is controlled by $Y$ separately.

$$\tilde{q} \equiv (-1 + q_1, q_3, -1 + q_2, q_4) \tag{28}$$

In the stable state, the expected revenues of $X$ and $Y$ are

$$u_X = \frac{v \cdot U_X}{v \cdot 1} = \frac{D(p, q, U_X)}{D(p, q, 1)}, \tag{29}$$

$$u_Y = \frac{v \cdot U_Y}{v \cdot 1} = \frac{D(p, q, U_Y)}{D(p, q, 1)}, \tag{30}$$

where 1 is the all-one vector, and the denominator normalizes the sum of the elements in $v$ to 1.

Since $u$ is linearly dependent on $U$, we have

$$\alpha u_X + \beta u_Y + \gamma = \frac{D(p, q, \alpha U_X + \beta U_Y + \gamma 1)}{D(p, q, 1)}. \tag{31}$$

If the strategy selected by $X$ satisfies $\tilde{p} = \alpha U_X + \beta U_Y + \gamma 1$, or the strategy selected by $Y$ satisfies $\tilde{q} = \alpha U_X + \beta U_Y + \gamma 1$, then $\alpha u_X + \beta u_Y + \gamma = 0$. This is the ZD strategy. Different sub-strategies can be obtained if parameters $\alpha$ and $\beta$ have different values. There are usually two types of sub-strategies of the ZD strategy which are the set strategy and the extortionate strategy. Under the set strategy, a player can unilaterally set the other's revenue to a fixed value. Under the extortionate strategy, the player choosing the extortionate strategy will receive a higher revenue than the other party, regardless of the other's strategy.

1. Set strategy

Under the set strategy, $X$ plays a game with $Y$, following the ZD strategy with $\alpha = 0$. By adjusting the cooperation probability, $X$ can unilaterally control the revenue of $Y$. In this case, $\tilde{p} = \beta U_Y + \gamma 1$, *i.e.*,

$$\tilde{p} = \begin{bmatrix} -1 + p_1 \\ -1 + p_2 \\ p_3 \\ p_4 \end{bmatrix} = \begin{bmatrix} \beta R + \gamma \\ \beta T + \gamma \\ \beta S + \gamma \\ \beta P + \gamma \end{bmatrix}. \tag{32}$$

Eliminate $\beta$ and $\gamma$, and represent $p_2$ and $p_3$ as $p_1$ and $p_4$.

$$u_Y = \frac{(1 - p_1)P + p_4 R}{(1 - p_1) + p_4} \tag{33}$$

In the mining dilemma, $T > R > P > S$. Thus, the revenue range of $Y$ is $P \leq u_Y \leq R$. When $p_1 \to 1$, $u_Y \to R$; when $p_4 \to 0$, $u_Y \to P$. Regardless of $Y$'s strategy, $X$ can set the long-term revenue of $Y$ to a fixed value.

2. Extortionate strategy

If $X$ plays a game with $Y$, following the ZD strategy with $-\beta/\alpha = \chi$ and $\gamma/\alpha = -(1 - \chi)P$, his/her revenue minus $Y$'s defection revenue $P$ will be times $Y$'s revenue minus $P$.

$$U_X - P1 = \chi(U_Y - P1) \tag{34}$$

$\chi$ is the extortion factor, $\chi \geq 1$. If the extortion factor is fixed as a constant, $X$ will continuously receive a high revenue, but $Y$ will not be encouraged to choose cooperation. Therefore, this paper sets the extortion factor as a dynamic factor $\chi = 10/P_C$, where $P_C$ is the cooperation probability. It can be observed that a small cooperation probability corresponds to a large extortion factor. In this case, each pool aims to receive a high revenue.

Suppose

$$\tilde{p} = \phi[(U_X - P1) - \chi(U_Y - P1)] = \begin{bmatrix} \phi[(R-P) - \chi(R-P)] \\ \phi[(S-P) - \chi(T-P)] \\ \phi[(T-P) - \chi(S-P)] \\ \phi[(P-P) - \chi(P-P)] \end{bmatrix} = \begin{bmatrix} -1 + p_1 \\ -1 + p_2 \\ p_3 \\ p_4 \end{bmatrix}. \tag{35}$$

Since $p_1, p_2, p_3, p_4 \in [0, 1]$, we have

$$0 < \phi \leq \frac{P - S}{(P - S) + \chi(T - P)}. \tag{36}$$

If $X$ adopts the extortionate ZD strategy, his/her revenue will depend on the strategy $q$ of $Y$. If $Y$ chooses the AllC strategy, *i.e.*, $q = (1, 1, 1, 1)$, then both $X$ and $Y$ will receive the maximum revenue. In this case, the revenues of $X$ and $Y$ can be respectively calculated by

$$u_X = \frac{P(T - R) + \chi[R(T - S) - P(T - R)]}{(T - R) + \chi(R - S)}, u_Y = \frac{R(T - S) + P(\chi - 1)(R - S)}{(T - R) + \chi(R - S)}. \tag{37}$$

3. Adaptive ZD strategy

Replace $R$ in the extortionate strategy formula.

$$U_X - V1 = \chi(U_Y - V1), \tag{38}$$

where $V$ is the reference revenue variable that adjusts the surplus, $P \leq V \leq R$. The adaptive zero-determinant strategy continuously adjust the value of $V$ through the game according to the environment. During the donation game model, a player choosing cooperation provide a revenue $b$ to his/her opponent at the cost $c$, $b > c > 0$. Hence, the reference revenue variable $V$ can be represented as

$$V = \sigma(R - P) + P = \sigma(b - c), \tag{39}$$

where, $\sigma \in [0, 1]$. Through the game process, the adaptive ZD strategy continuously adjusts the $V$ value, which equals the continuous adjustment of $\sigma$. The change of parameter $\sigma$ determines whether the ZD strategy adopter is a selfish extortioner or generous donator.

This paper proposes a adaptive Strategy allowing the player to dynamically set the corporation probability of the next round based on its previous revenues. Suppose that, in the $t^{th}$ round, the revenue of $X$ is $u^{(t)}$, the cooperation probability is $p^{(t)}$, and the average

revenue of the previous rounds is $\bar{u}^{(t)}$. Then, if $u^{(t)} \geq \bar{u}^{(t)}$, $X$ will choose cooperation in this round, and the probability of cooperation in the next round is adjusted to

$$p^{(t+1)} = p^{(t)} + \frac{1 - p^{(t)}}{1 + e^{\theta(t)}}, \tag{40}$$

where

$$\theta(t) = \sum_{i=1}^{t} p^{(t)} \left( u^{(i)} - \bar{u}^{(t)} \right). \tag{41}$$

Obviously, when $\theta(t) < 0$, if $\theta(t)$ changes 0.1 unit each time, the adjustment amount of cooperation probability is greater than that when $\theta(t) > 0$.

If $u^{(t)} < \bar{u}^{(t)}$, $X$ will choose defection in this round, and the probability of cooperation in the next round is adjusted to

$$p^{(t+1)} = p^{(t)} - \frac{1 - p^{(t)}}{1 + e^{\theta(t)}}. \tag{42}$$

# EXPERIMENTS AND RESULTS ANALYSIS

Experiments were carried out to verify the revenue changes of different game strategies during the BWH attacks between mining pools.

## Experimental setting

The following are some classic strategies for the prisoner's dilemma model.

1. Always Cooperate (AllC): always choose cooperation, regardless of the opponent's behaviors. In this case, the selection probability is $AllC = (1, 1, 1, 1)$, i.e., the BWH attack will never be launched, and the infiltration rate $r = 0$.

2. Always Defect (AllD): always choose defection, regardless of the opponent's behaviors. In this case, the selection probability is $AllD = (0, 0, 0, 0)$, i.e., the BWH attack will always be launched, and the infiltration rate $r > 0$.

3. Tit For Tat (TFT): choose cooperation initially, and choose the same strategy as the opponent's strategy in the previous round. In this case, the selection probability is $TFT = (1, 0, 1, 0)$.

4. Win-Stay, Lose-Shift (WSLS): set a threshold for revenue, and choose cooperation in the first round; in each of the following rounds, if the revenue is above the threshold, keep the strategy; otherwise, choose the opposite strategy. Here, the selection probability is defined as $WSLS = (1, 0, 0, 1)$.

5. Random: choose a random strategy at a discrete probability.

$X$ has a total of eight strategies, which are AllC, AllD, TFT, WSLS, Random, set strategy, extortionate strategy, and adaptive strategy. $Y$ has a total of five strategies, which are AllC, AllD, TFT, WSLS, and Random. Through orthogonal design, a total of 30 different games were obtained for our experiments.

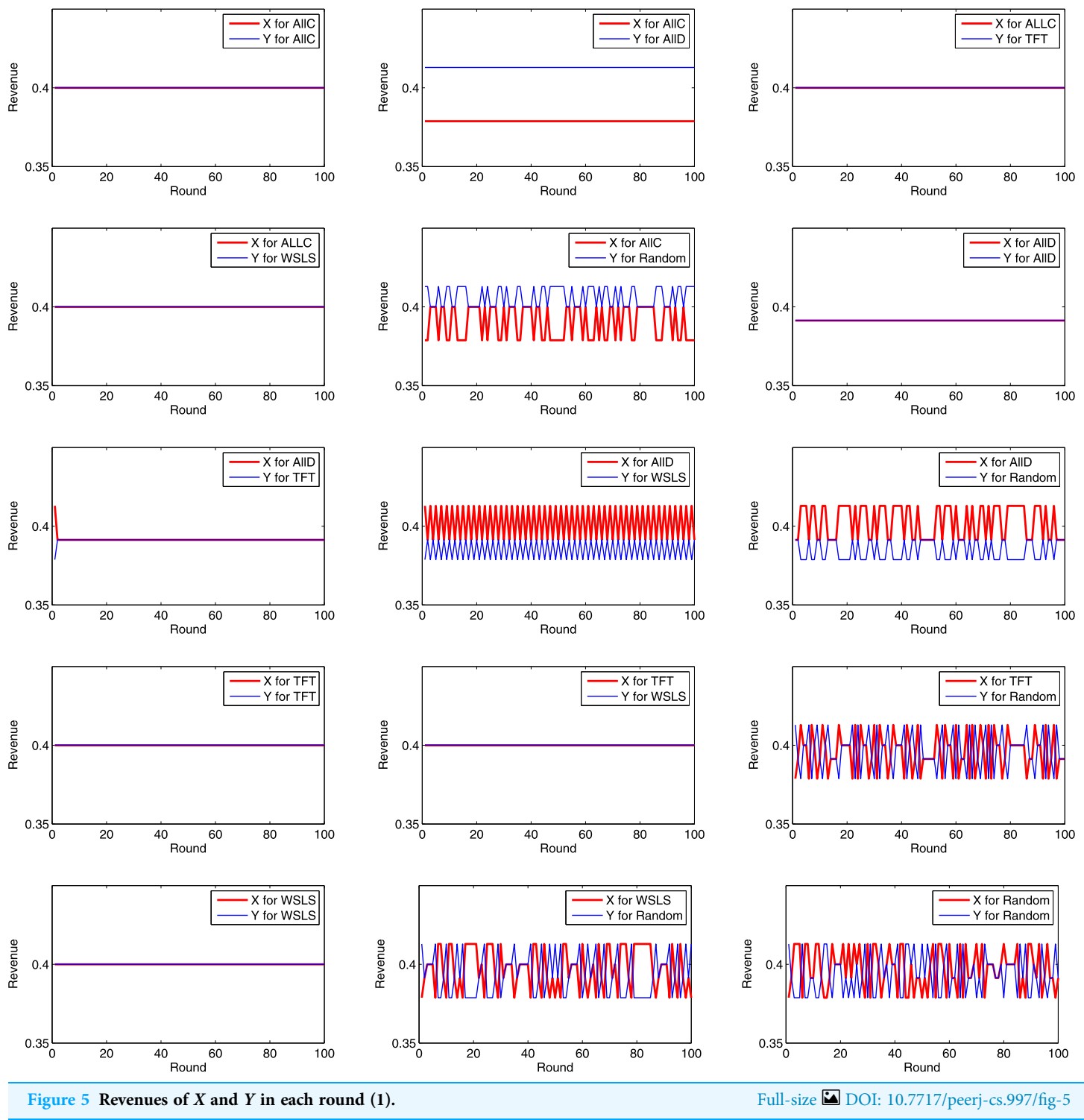

**Figure 5 Revenues of *X* and *Y* in each round (1).**     

## Simulation results and analysis

### Simulation results of pairwise game

Let $h_1 = h_2 = 0.4$, and $r_1 = r_2 = 0.1$. Each game was simulated for 100 rounds. Then, the revenues of *X* and *Y* in each round of each game are summarized as the following Figs. 5 and 6.

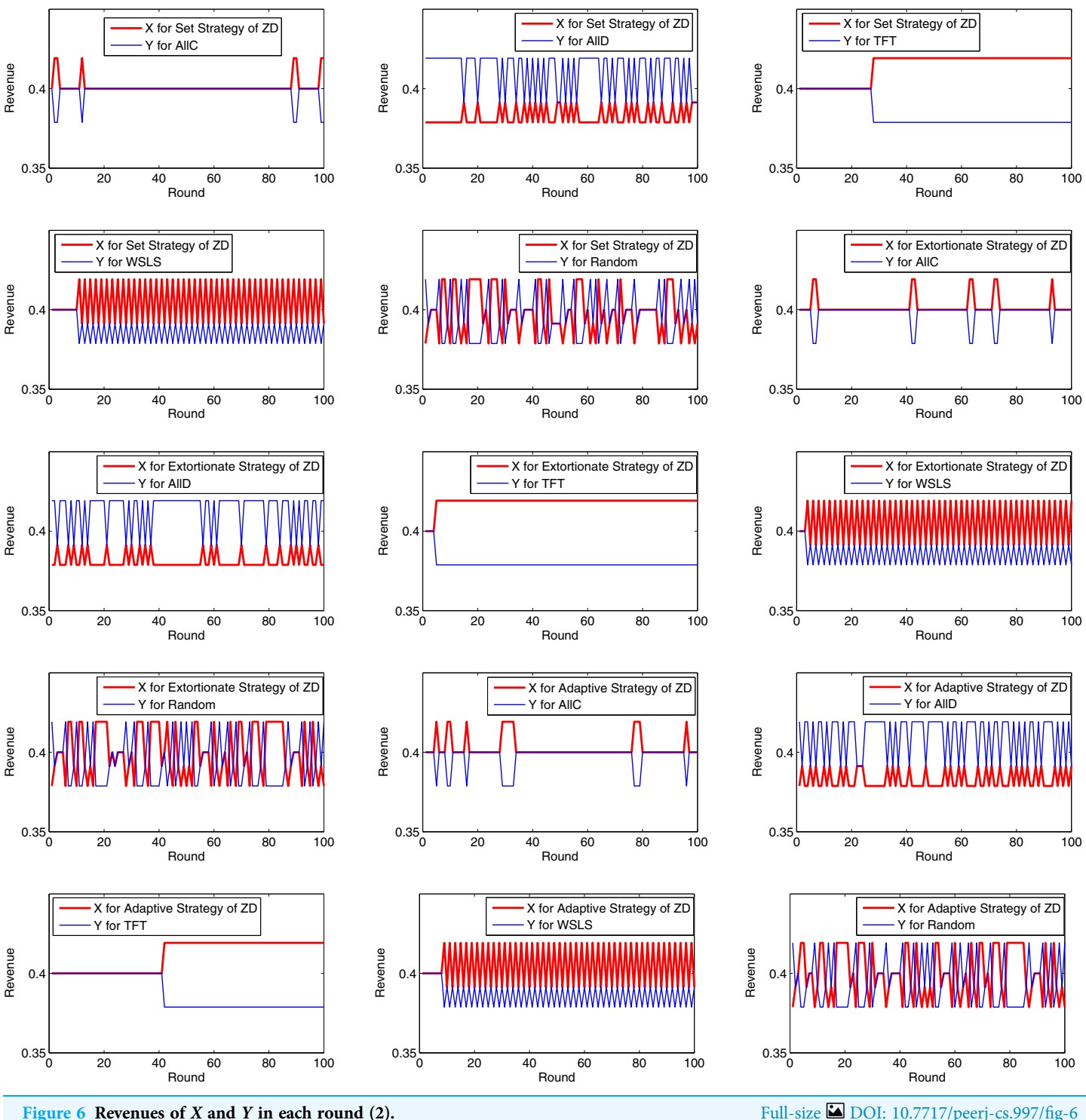

**Figure 6 Revenues of *X* and *Y* in each round (2).**

**Table 3  All revenues of X and Y.**

| | | Y | | | | | | | | | |
|---|---|---|---|---|---|---|---|---|---|---|---|
| | | AllC | | AllD | | TFT | | WSLS | | Random | |
| | | X's | Y's | X's | Y's | X's | Y's | X's | Y's | X's | Y's |
| X | AllC | 40 | 40 | 37.88 | 41.29 | 40 | 40 | 40 | 40 | 39.00 | 40.61 |
| | AllD | 41.29 | 37.88 | 39.13 | 39.13 | 39.15 | 39.12 | 40.21 | 38.51 | 40.27 | 38.47 |
| | TFT | 40 | 40 | 39.12 | 39.15 | 40 | 40 | 40 | 40 | 39.59 | 39.62 |
| | WSLS | 40 | 40 | 38.51 | 40.21 | 40 | 40 | 40 | 40 | 39.73 | 39.39 |
| | Random | 40.61 | 39.00 | 38.59 | 40.06 | 39.54 | 39.50 | 39.58 | 39.48 | 39.73 | 39.39 |
| | Set strategy | 40.13 | 39.85 | 38.26 | 41.08 | 41.40 | 38.45 | 40.47 | 38.65 | 39.78 | 39.70 |
| | Extortionate strategy | 40.17 | 39.81 | 38.15 | 41.31 | 41.84 | 37.96 | 40.52 | 38.54 | 39.99 | 39.47 |
| | Adaptive strategy | 40.25 | 39.74 | 38.49 | 40.98 | 41.13 | 38.75 | 40.48 | 38.67 | 39.96 | 39.51 |

**Table 4  Total revenues of X and Y.**

| | | Y | | | | |
|---|---|---|---|---|---|---|
| | | AllC | AllD | TFT | WSLS | Random |
| X | AllC | 80.00 | 79.17 | 80.00 | 80.00 | 79.61 |
| | AllD | 79.17 | 78.26 | 78.27 | 78.72 | 78.74 |
| | TFT | 80.00 | 78.27 | 80.00 | 80.00 | 79.21 |
| | WSLS | 80.00 | 78.72 | 80.00 | 80.00 | 79.12 |
| | Random | 79.61 | 78.65 | 79.04 | 79.06 | 79.12 |
| | Set strategy | 79.99 | 79.34 | 79.85 | 79.13 | 79.47 |
| | Extortionate strategy | 79.98 | 79.46 | 79.81 | 79.07 | 79.46 |
| | Adaptive strategy | 79.99 | 79.47 | 79.88 | 79.15 | 79.47 |

**Table 5  Average variance of X and Y.**

| | | AllC | | AllD | | TFT | | WSLS | | Random | |
|---|---|---|---|---|---|---|---|---|---|---|---|
| | | X's | Y's | X's | Y's | X's | Y's | X's | Y's | X's | Y's |
| X | Set Strategy | 4.56 | 5.57 | 3.34 | 16.62 | 3.38 | 4.12 | 18.17 | 5.35 | 22.28 | 21.43 |
| | Extortionate Strategy | 3.24 | 3.95 | 2.81 | 13.98 | 2.23 | 2.72 | 17.37 | 5.77 | 21.64 | 21.33 |
| | Adaptive Strategy | 4.98 | 6.07 | 3.64 | 18.12 | 4.08 | 4.97 | 18.40 | 5.11 | 21.68 | 21.17 |

It can be inferred from the above figures that if $X$ chooses AllC, and $Y$ chooses AllC, TFT, or WSLS, the two parties always cooperate with each other, and each party receives a revenue of $R$ in each round. If $X$ chooses AllC, and $Y$ chooses AllD, $X$ receives a revenue of $S$ in each round, while $Y$ receives $T$. Therefore, if $Y$ unilaterally launches a BWH attack, his/her own revenue will increase at the cost of the other's revenue. If $X$ chooses AllD, and $Y$ chooses AllD, the revenues of both $X$ and $Y$ are $P$. If $X$ chooses AllD, and $Y$ chooses TFT, the strategy will evolve into AllD in the second round. If $X$ chooses AllD,

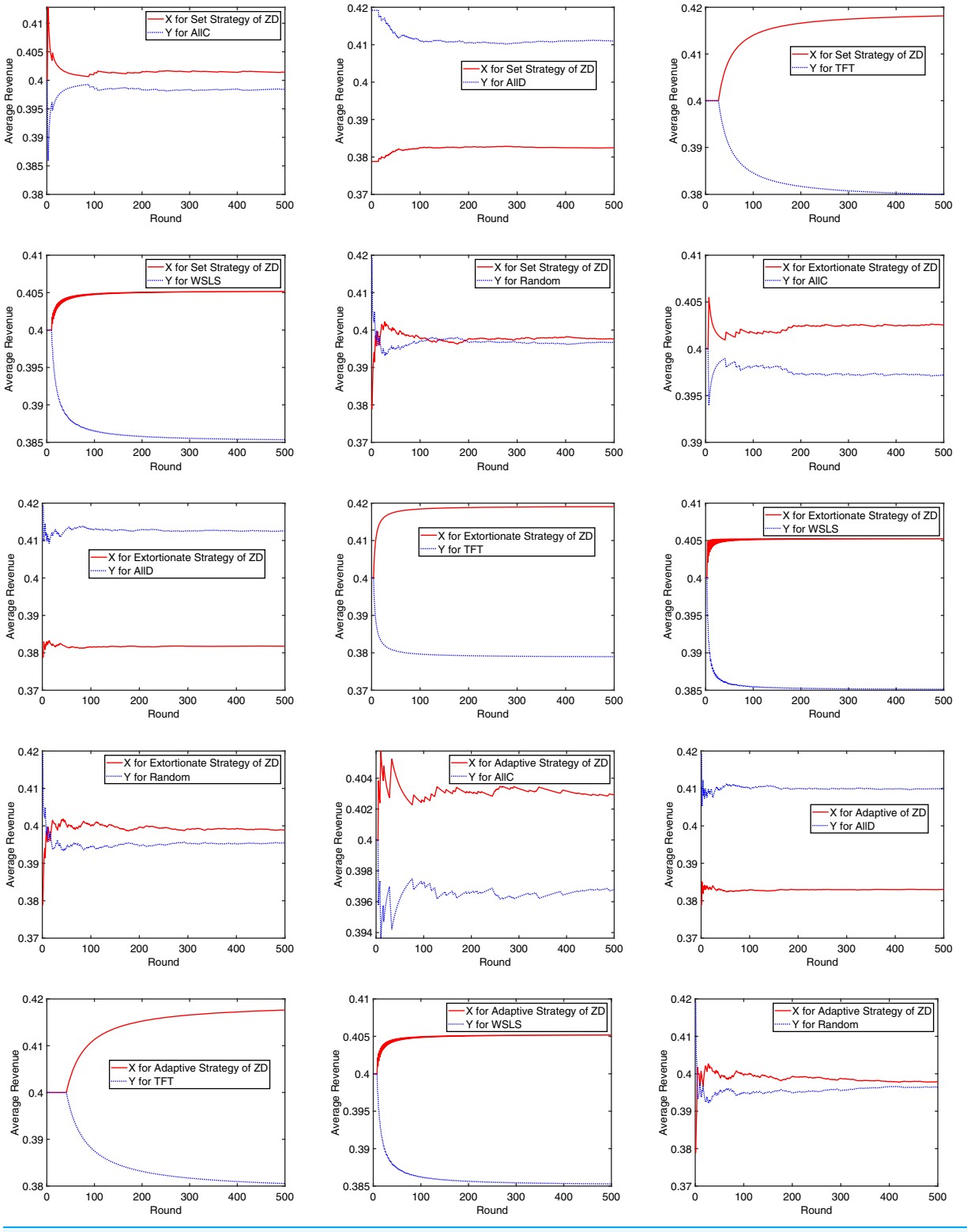

**Figure 7 Convergence of the average revenue.**

and $Y$ chooses WSLS, its strategy for each round will be the opposite to that for the previous round. If $X$ chooses TFT, and $Y$ chooses TFT or WSLS, the two parties will always cooperate with each other. If both choose WSLS, the two parties will always cooperate with each other.

### Analysis of game revenue

Tables 3 and 4 present all revenues of $X$ and $Y$ and their total revenues after 100 rounds of the game under different strategy combinations, respectively. As shown in Table 4, if $Y$ chooses AllD, the total revenues obtained by $X$ and $Y$ when $X$ chooses the three ZD strategies, are greater than those when $X$ chooses other five strategies, and the adaptive ZD strategy has the highest revenues; When $Y$ chooses AllC, TFT or WSLS and $X$ chooses the adaptive ZD strategy except AllC, TFT and WSLS, $X$ and $Y$ obtain the maximum total revenues; when $Y$ chooses Random and $X$ chooses the adaptive ZD strategy except AllC, $X$ and $Y$ can also obtain the maximum total revenues. It is a strong indication that when $X$ adopts the three ZD strategies, $X$ can not only increase his/her own revenue, but also push up the revenue of the entire system, thereby mitigating the opponent's attack.

The paper repeats the games 50 times that $X$ chooses one of the three ZD Strategies, and $Y$ chooses one of the other five Strategies. The average variance of $X$ and $Y$ in each game is shown in Table 5. As can be seen, all the variances are very small, which means that the revenues of $X$ and $Y$ in all games are very stable.

### Analysis of convergence of average revenue

Next, the initial cooperation probability of $X$ was set to 0.9, and 500 rounds of game were simulated under the set strategy, extortionate strategy, and adaptive strategy of the ZD, respectively. The average revenue variation of each party is recorded in Fig. 7. Obviously, every set of game strategies could converge very quickly. After less than 10 rounds at least or no more than 40 rounds at most, the average revenue of both $X$ and $Y$ has been basically stable, indicating that the mining dilemma can be effectively mitigated.

## CONCLUSIONS

This paper mainly studied how to mitigate the mining dilemma of block withholding attack between the mining pools by means of zero-determinant strategies. It deduced the calculation formula for the actual revenue of the mining pool at first when the block withholding attack is launched. And then, the ZD strategies such as the set strategy and the extortionate strategy are theoretically studied to solve the Nash equilibrium problem of the mining dilemma. Based on these theories, the adaptive ZD strategy was put forward, changing the corporation probability of the next round based on the previous revenues. Finally, 30 sets of game strategies were selected and simulated to show the actual revenue variation of the pools. The experimental simulation indicated that these ZD strategies, especially the proposed adaptive strategy, can promote the cooperation between the pools and increase both the overall revenue of the pool and the revenue of each miner. However, the paper only considered the two-player applying discrete strategy. Therefore, the authors will continue to study the multi-player applying discrete strategy iterative game, the two-player and multi-player applying continuous strategy games.

### Funding

This work was supported by the National Natural Science Foundation of China (No. 61502151) and the Shandong Key R&D (Major Scientific and Technological Innovation) Project of China (No. 2021CXGC010108). There was no additional external funding received for this study. The funders had no role in study design, data collection and analysis, decision to publish, or preparation of the manuscript.

### Grant Disclosures

The following grant information was disclosed by the authors:
National Natural Science Foundation of China: 61502151.
Shandong Key R&D (Major Scientific and Technological Innovation) Project of China: 2021CXGC010108.

### Competing Interests

The authors declare that they have no competing interests.

### Author Contributions

- Min Ren conceived and designed the experiments, performed the computation work, prepared figures and/or tables, authored or reviewed drafts of the article, and approved the final draft.
- Hongfeng Guo analyzed the data, authored or reviewed drafts of the article, and approved the final draft.
- Zhihao Wang performed the experiments, prepared figures and/or tables, and approved the final draft.

### Data Availability

   The raw data and code are available in the Supplemental Files.

### Supplemental Information

Supplemental information for this article can be found online at http://dx.doi.org/10.7717/peerj-cs.997#supplemental-information.

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
