# Peer review of "Mitigation of block withholding attack based on zero-determinant strategy"

_PeerJ Computer Science, doi:10.7717/peerj-cs.997_

## Round 0.1 · original submission · Major Revisions

Please carefully revise the manuscript based on the reviewer's comments. Kindly check the language to omit any typos and grammatical mistakes.

Reviewer 1 ·

Basic reporting

I recommend to (also) display the revenue of the independent miners in an additional Figure 4(c).

Could mention, at least as outlook / room for further consideration:
Would mining difficulty adjustment (after 14 days) adjust pool revenue facing a block withholding attack?

Section "ZD strategy" (from 7/17, line 225) actually lacks a precise and compact definition of a "ZD strategy":
Lines 245/246 on page 8/17 do provide some conditions ("If the strategy selected by X satisfies ... This is the ZD strategy"), yet unclear what set of candidate "strategies" (which then do or do not fulfil the conditions) X can select from.

Conclusions sections very short. Should also summarize restrictions (e.g.: for the case of two pools) and outline further work.

Should define all abbreviations on first mention (TFT = Tit for tat? WSLS=?)

Should use a small space (typically "\," in LaTeX) between number and unit (10mins ->10 mins, 300BTC -> 300 BTC).
"Pool1"/"Pool2" should use subscript für index values ("Pool$_1$").

Experimental design

Usual not to find experiments replicated.
Would expect multiple runs of 100 (or 500) steps each and variance reviewed.
If not, should (at least) argue why - according to the authors - this is not necessary.

Validity of the findings

no comment

Additional comments

no (furher) comment

Reviewer 2 ·

Basic reporting

This paper deduces the calculation formula of the actual income of the mining pool when the block withholding attack is launched. The ensemble strategy, extortion strategy and the proposed adaptive strategy of ZD are studied. Then, the mitigation of block interception attack based on ensemble strategy, extortion strategy and zero-determinant adaptive strategy is theoretically studied. The authors also selected and simulated 30 sets of game strategies to illustrate changes in the pool's returns.

Experimental design

Although this paper is structurally complete and logically clear, there are a few problems in the manuscript, which needs the author to modify for being accepted. The problems in the manuscript are shown below in detail.
1. In the abstract section, the author should make clear which works are his own, rather than linearly describe the prose structure of the work.
2. Some formulas are not in the same format as (11) (15). Authors should proofread carefully to ensure that all formulas are in the same format.
3. In some of the figures, it is difficult to see the specific situation corresponding to Y as shown in Figure 6 and Figure 7. The author should make corresponding changes to make the figure clearer.
4. In the conclusion part, the author should make a concise summary of each chapter in the conclusion, clarifying his own contribution.
5. The following major workings are missing in the references: Multiple cloud storage mechanism based on blockchain in smart homes; Novel vote scheme for decision-making feedback based on blockchain in internet of vehicles.

Validity of the findings

Although this paper is structurally complete and logically clear, there are a few problems in the manuscript, which needs the author to modify for being accepted. The problems in the manuscript are shown below in detail.
1. In the abstract section, the author should make clear which works are his own, rather than linearly describe the prose structure of the work.
2. Some formulas are not in the same format as (11) (15). Authors should proofread carefully to ensure that all formulas are in the same format.
3. In some of the figures, it is difficult to see the specific situation corresponding to Y as shown in Figure 6 and Figure 7. The author should make corresponding changes to make the figure clearer.
4. In the conclusion part, the author should make a concise summary of each chapter in the conclusion, clarifying his own contribution.
5. The following major workings are missing in the references: Multiple cloud storage mechanism based on blockchain in smart homes; Novel vote scheme for decision-making feedback based on blockchain in internet of vehicles.

Reviewer 3 ·

Basic reporting

In this manuscript, the authors investigated block withholding attack in blockchain based on game theory. They consider the situation that only two mining pools exist and one or both pools attack each other. By calculating revenue of each mining pool, they pointed out that choosing whether it attacks the other pool or not is similar to choice of action in the prisoner's dilemma game. Then, they investigated zero-determinant (ZD) strategies in the iterated prisoner's dilemma (IPD) game, and showed that an adaptive ZD strategy promotes cooperation between pools.

Although their idea about applying ZD strategies to blockchain mining is interesting, I think that the current version should not be published. The reason is that, whereas the first part "Mining dilemma analysis" is significant as an application of game theory, the second part "ZD strategy" does not contain any novel results. They just investigated ZD strategies in IPD, which have already been studied by many researchers, and the paper does not advance our understanding at all.

I think that the problem comes from the fact that the authors reduced their model to IPD, which is too simple, although action in their original model is the continuous parameter $r_i$. I recommend that the authors investigate ZD strategies in their mining game without reducing it to IPD. In research on ZD strategies, McAvoy and Hauert extended ZD strategies to continuous action space [PNAS 113(13), 3573 (2016)]. I think that this paper is useful for improving their manuscript.

In addition, as a minor comment, their survey on ZD strategies is not sufficient.
There are many papers which should be mentioned in "Related works". For example,
*ZD alliance [Hilbe, Wu, Traulsen, Nowak, PNAS 111(46), 16425 (2014)]
*ZD strategies in noisy games [Hao, Rong, Zhou, Phys. Rev. E 91, 052803 (2015)]
*ZD strategies in games with a discount factor [Hilbe, Traulsen, Sigmund, Games and Economic Behavior 92, 41 (2015)]
*Extension to alternating games [McAvoy, Hauert, Theoretical Population Biology 113, 13 (2017)]
*Consistency and independence of ZD strategies [Ueda, Tanaka, PLoS ONE 15(4), e0230973 (2020)]
*Extension to memory-$n$ strategies [Ueda, Royal Society Open Science 8, 202186 (2021)]

I hope that the authors resubmit a revised version.

Experimental design

no comment

Validity of the findings

no comment

---

## Round 0.2 · accepted · Accept

Thank you for revising the manuscript based on the reviewers comments. The reviewers are satisfied with the revised manuscript and recommend acceptance.

Reviewer 2 ·

Basic reporting

Revised paper is much improved and can be accepted now.

Experimental design

Revised paper is much improved and can be accepted now.

Validity of the findings

Revised paper is much improved and can be accepted now.

Reviewer 3 ·

Basic reporting

The authors sincerely addressed my comments or explained the novelty of their work in their rebuttal. Now I understand that the innovation of their work is an application of zero-determinant strategies to block withholding attack, and recommend an acceptance.

Experimental design

no comment

Validity of the findings

no comment